TOPICAL REVIEW

# The impact of androgens on cardiovascular control mechanisms in polycystic ovary syndrome: Recent advances and translational approaches

Zoe H. Adams[1],# (ID), Danielle E. Berbrier[2],# (ID), Brittany K. Schwende[2], Will Huckins[2] (ID), Cory T. Richards[1] (ID), D. Aled Rees[3,4], Charlotte W. Usselman[2] and Rachel N. Lord[1] (ID)

[1]*Cardiff School of Sport and Health Sciences, Cardiff Metropolitan University, Cardiff, UK*
[2]*Department of Kinesiology and Physical Education, McGill University, Montreal, Canada*
[3]*Cardiff and Vale University Health Board, Cardiff, UK*
[4]*Neuroscience and Mental Health Institute, School of Medicine, Cardiff University, Cardiff, UK*

Handling Editors: Laura Bennet & Christopher Lear

The peer review history is available in the Supporting Information section of this article (https://doi.org/10.1113/JP287288#support-information-section).

**Abstract figure legend** The cardiovascular control mechanisms that may be altered in individuals with polycystic ovary syndrome (PCOS). Pink boxes indicate mechanisms for which there is evidence in individuals with PCOS; blue boxes indicate putative mechanisms with no current data available in human PCOS cohorts. SNA, sympathetic nerve activity; NO; nitric oxide.

#Denotes joint first authorship.

**Abstract** Polycystic ovary syndrome (PCOS) is the most common endocrinopathy in pre-menopausal females. The condition is associated with an increased prevalence of cardiovascular risk factors, including hypertension. Observational studies report that some blood pressure control mechanisms are altered in PCOS compared to controls (sympathetic nervous system activity, endothelial and vasodilator function, renin angiotensin aldosterone system activation), and that these impairments correlate with androgen hormone levels, which are chronically elevated in the condition. As such, hyperandrogenism is the proposed locus of origin for the link between PCOS and cardiovascular dysfunction, yet the underlying mechanisms remain poorly understood. Preclinical work has provided some insight into how androgens modulate blood pressure control in PCOS. However there are marked discrepancies between the effects of androgens in cellular and tissue studies *versus in vivo* animal and human PCOS studies. This may be related to the heterogeneity of the preclinical models and samples used in this research and whether preclinical work is modelling hyperandrogenism in physiologically relevant terms for PCOS. This review collates preclinical and clinical evidence to summarise what is known and what remains unknown about cardiovascular control mechanisms in PCOS. We examine aspects of blood pressure regulation that are altered in other hypertensive cohorts, presenting current evidence for a mechanistic role of androgens on these systems, while acknowledging the diverse experimental models and participant cohorts from which the results are derived.

(Received 19 November 2024; accepted after revision 24 March 2025; first published online 27 April 2025)

**Corresponding author** Z. Adams: Cardiff School of Sport and Health Sciences, Cardiff Metropolitan University, Cardiff, UK. Email: zadams@cardiffmet.ac.uk

## Introduction

Polycystic ovary syndrome (PCOS) is the most common endocrine disorder among reproductive-aged females, with a prevalence between 6% and 20% (Chiaffarino et al., 2022; Deswal et al., 2020; Teede et al., 2023). The most commonly cited diagnostic criteria for PCOS are the 2003 Rotterdam criteria (Rotterdam, 2004), which define PCOS based on the presence of at least two of three characteristics: clinical (i.e. hirsutism, acne, alopecia) and/or biochemical (i.e. elevated serum androgens) hyperandrogenism, oligo- or anovulation and morphological polycystic ovaries on ultrasound, although some guidelines now recommend elevated anti-Mullerian hormone as an alternative marker of polycystic ovaries (Teede et al., 2023). The hyperandrogenic PCOS phenotype is present in 65%–90% of all PCOS cases (Chiaffarino et al., 2022; Kanbour & Dobs, 2022) and is characterised by elevated androstenedione (occurs in 88% of cases), total testosterone (65%) and/or dehydroepiandrosterone sulphate (DHEAS; 30%) (Kanbour & Dobs, 2022), alongside reduced sex hormone binding globulin (SHBG) (Deswal et al., 2018). Importantly PCOS is associated with an elevated risk of cardiovascular disease (Berni et al., 2021; Holte et al., 1996; Vine et al., 2024). This risk is attributed to higher prevalence of cardiovascular risk factors in PCOS, including elevated blood pressure (Holte et al., 1996; Joham et al., 2015), sympathetic nerve activity (Sverrisdóttir et al., 2008), endothelial dysfunction (Berbrier et al., 2023; Sprung et al., 2013), atherosclerosis (Talbott et al., 2000), obesity (Orio et al., 2016), insulin resistance (Carmina et al., 2005) and dyslipidaemia (Wild et al., 2011). Crucially, cardiovascular risk factors and cardiovascular disease in PCOS have both been linked to hyperandrogenism, even when body mass index (BMI) is accounted for (Berbrier et al., 2023; Hirschberg, 2023; Sverrisdóttir et al., 2008).

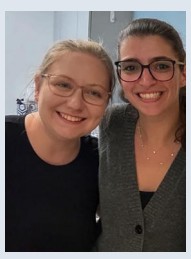

**Zoe Adams** (left) is a postdoctoral researcher at Cardiff Metropolitan University School of Sport and Health Sciences, working with Dr Rachel Lord and Prof Aled Rees. She is interested in the integrative mechanisms driving the development of hypertension in females, especially the interaction of sex hormones and autonomic blood pressure control. **Danielle Berbrier** (right) is a PhD candidate in the Department of Kinesiology and Physical Education at McGill University, under the supervision of Dr. Charlotte Usselman. Danielle's current research focuses on investigating skeletal muscle circulation as a site of cardiovascular dysfunction in females with polycystic ovary syndrome.

The relative risk of hypertension in PCOS is almost two-fold greater than that of controls (Amiri et al., 2020). This increased risk in PCOS is particularly notable given that prevalence of hypertension in premenopausal females is generally low compared to age-matched males and only increases substantially with age after menopause (Benjamin et al., 2017; Burt et al., 1995; Gu et al., 2002). Given that PCOS is associated with elevated blood pressure well before menopause (Mellembakken et al., 2021; Wekker et al., 2020), blood pressure regulation may be altered in individuals with PCOS.

To understand the mechanisms by which androgens influence blood pressure regulation in PCOS, researchers have drawn upon the substantial body of preclinical research investigating the vascular actions of androgens. This work suggests that androgens have numerous vascular effects (Lorigo et al., 2020; Lucas-Herald & Touyz, 2022; Reckelhoff, 2019) via both genomic and non-genomic pathways (Lucas-Herald et al., 2017). However there are still significant gaps in our understanding of how hyperandrogenism may alter blood pressure regulation in PCOS. For instance, there exist some marked discrepancies between the effects of androgens in cellular and tissue studies *versus in vivo* animal studies *versus* human PCOS research. The animal models, tissue samples and cell cultures used in this preclinical research vary in terms of sex and androgen exposure (i.e. duration, concentration, exogenous *versus* endogenous). Importantly caution is warranted when extrapolating findings from male disease models and testosterone concentrations that are supraphysiological or physiological for males, as there is evidence that androgens exert sex-specific effects on metabolic risk (Ruth et al., 2020), meaning that these data may not directly translate to the pathophysiology of PCOS. Additionally, animal models of PCOS exhibit significant heterogeneity in the characteristics of the condition that they display, which may depend on the mechanism of androgen exposure (i.e. prenatal *versus* postnatal, which androgen used) (see Padmanabhan & Veiga-Lopez, 2013a; Shi & Vine, 2012; Walters et al., 2018 for reviews). As such, it may be that some preclinical studies are more physiologically relevant to human PCOS than others. Therefore, this review aims to summarise the available evidence for a mechanistic role of androgens in cardiovascular regulation in PCOS, while highlighting the participant cohorts and preclinical models from which the data are derived (Table 2). We discuss the cardiovascular control mechanisms that contribute to hypertension in other patient groups (Fig. 1; Table 1) and present available clinical and preclinical evidence for the action of androgens at each of these levels. We have prioritised data from individuals with PCOS or female models of chronic androgen exposure, where available. When discussing human studies, we have used gender-inclusive language by referring to participants diagnosed with the condition as individuals with PCOS.

## Androgens and blood pressure in PCOS

*Clinical populations.* A moderate amount of evidence now suggests that individuals with PCOS have elevated blood pressure and a higher prevalence of hypertension. That is, multiple cross-sectional studies have reported that blood pressure is elevated in individuals with PCOS relative to controls (Fu et al., 2023; Holte et al., 1996; Luque-Ramírez et al., 2014; Mellembakken et al., 2021). Furthermore, several studies have reported an increased hypertension prevalence in individuals with PCOS (5.5%–11.8%) *versus* control participants ($\leq$2%) (Joham et al., 2015; Mellembakken et al., 2021), whereas two independent meta-analyses have concluded that PCOS is associated with a ~1.7-fold increased relative risk of hypertension *versus* controls (Amiri et al., 2020; Wekker et al., 2020). Although group average blood pressure values do not typically reach hypertensive levels (i.e. European guidelines of >140/90 mmHg (McEvoy et al., 2024)) in these studies, high-normal blood pressures still represent a risk factor for further cardiovascular disease in PCOS, given that cardiovascular risk increases with increasing systolic blood pressure above 115 mmHg (Lewington et al., 2002). Furthermore, recent work has shown that blood pressure is associated with cardiovascular risk at lower thresholds (10–20 mmHg lower) in females than in males (Ji et al., 2021; Wang et al., 2022). As such high-normal blood pressure and increased hypertension prevalence are likely contributors to the higher cardiovascular risk in PCOS.

Whether the elevated hypertension risk is attributed to hyperandrogenism or to other cardiovascular risk factors in PCOS (e.g. insulin resistance, obesity) remains debated. In a large cohort study ($n = 151$) of PCOS patients, Chen and colleagues (2007) demonstrated that both the free androgen index and total testosterone predicted resting blood pressure independently of age and BMI (Chen et al., 2007). However, another study of similar sample size ($n = 153$) reported that ambulatory blood pressure was similar between individuals with hyperandrogenic and non-hyperandrogenic PCOS (Franik et al., 2021). Interestingly, a smaller study ($n = 60$) observed that blood pressure was higher in obese individuals with PCOS than in BMI-matched controls, but lower in lean individuals with PCOS than in lean controls, which suggests that obesity drives hypertension in PCOS (Perusquía et al., 2023). However this finding is by no means ubiquitous, as elevated blood pressures and hypertension have been reported in lean individuals with PCOS (Mellembakken et al., 2021), and a recent meta-analysis demonstrated similar blood pressures between lean and

**Table 1. Summary of the cardiovascular risk factors in polycystic ovary syndrome (PCOS) compared to healthy controls (CTRL), highlighting relationships between risk factors and body mass index (BMI), androgens and insulin resistance.**

| Cardiovascular risk factor | Magnitude of difference in PCOS *vs.* CTRL | Relationship with BMI | Relationship with androgens | Relationship with insulin resistance markers |
|---|---|---|---|---|
| **Hypertension prevalence** | ↑ ~ **1.7 × relative risk** (Amiri et al., 2020; Wekker et al., 2020) | | | |
| **Blood pressure** | ↑ **3–8 mmHg** (Fu et al., 2023; Holte et al., 1996; Luque-Ramírez et al., 2014) | **Lean:** PCOS > CTRL (Mellembakken et al., 2021) PCOS < CTRL (Perusquía et al., 2023) **Overweight/obese:** PCOS > CTRL (Perusquía et al., 2023) **Within PCOS:** Lean < obese (Ketel et al., 2010; Perusquía et al., 2023) | **All BMI:** Positive correlation (Chen et al., 2007; Holte et al., 1996) **Lean:** Positive correlation (Mellembakken et al., 2021) | **All BMI:** Positive correlation (Chen et al., 2007; Holte et al., 1996) **Lean:** Positive correlation (Mellembakken et al., 2021) |
| **MSNA** | ↑ **8–15 bursts/100 HB 6–10 burst/min** (Lambert et al., 2015; Shorakae et al., 2018; Sverrisdóttir et al., 2008) | **Lean:** PCOS > CTRL (Sverrisdóttir et al., 2008) **Overweight/obese:** PCOS > CTRL (Lambert et al., 2015; Shorakae et al., 2018) | **Lean:** Positive correlation (Sverrisdóttir et al., 2008) **Overweight/obese:** No correlation (Lambert et al., 2015) | **Lean:** No correlation (Sverrisdóttir et al., 2008) **Overweight/obese:** Positive correlation (Lambert et al., 2015) |
| **Endothelial function (FMD)** | ↓ **3.4%** (Sprung et al., 2013) | **Lean:** PCOS < CTRL (Berbrier et al., 2023; Cussons et al., 2009; Orio et al., 2004; Sorensen et al., 2006; Soyman et al., 2011; Tarkun et al., 2004) **Overweight/obese:** PCOS < CTRL (Cascella et al., 2008; Diamanti-Kandarakis et al., 2006; Kravariti et al., 2005; Meyer et al., 2005; Sprung et al., 2014) **Within PCOS:** Lean = Overweight/obese (Kravariti et al., 2005; Sprung et al., 2014) | **Lean:** Negative correlation (Berbrier et al., 2023; Carmina et al., 2005; Sorensen et al., 2006) **Overweight:** Negative correlation (Kravariti et al., 2005) **Obese:** Negative correlation (Meyer et al., 2005) | **Lean:** Negative correlation (Orio et al., 2004; Soyman et al., 2011; Tarkun et al., 2004) **Overweight:** Negative correlation (Kravariti et al., 2005) |

(*Continued*)

**Table 1. (Continued)**

| Cardiovascular risk factor | Magnitude of difference in PCOS *vs.* CTRL | Relationship with BMI | Relationship with androgens | Relationship with insulin resistance markers |
|---|---|---|---|---|
| **Arterial stiffness (e.g. PWV)** | ↑ **0.38–0.8 m/s** (Kelly et al., 2002; Meyer et al., 2005; Sun et al., 2022) | **Lean:** PCOS > CTRL (Kilic et al., 2021; Soares et al., 2009) **Overweight/obese:** PCOS > CTRL (Kelly et al., 2002; Meyer et al., 2005) | **Lean:** Positive correlation (Kilic et al., 2021). **Overweight/obese:** Positive correlation (Burlá et al., 2019) | **Lean:** Positive correlation (Abacioglu et al., 2021; Sasaki et al., 2011) **Overweight/obese :** Positive correlation (Meyer et al., 2005) |
| **Atherosclerosis (CIMT)** | ↑ **0.072 mm** (Meyer et al., 2012) | **Lean:** PCOS > CTRL (Orio et al., 2004) **Overweight/obese:** PCOS > CTRL (Luque-Ramírez et al., 2007) **Within PCOS:** Lean = Overweight/obese (Ketel et al., 2010; Luque-Ramírez et al., 2007) | **Lean:** Positive correlation (Orio et al., 2004) **Overweight/obese:** Positive correlation (Luque-Ramírez et al., 2007) | **Lean:** Positive correlation (Carmina et al., 2005) **Overweight/obese:** Positive correlation (Luque-Ramírez et al., 2007) |

*Note:* Studies finding no effect of PCOS on these risk factors can be found in the main body of the text.

Abbreviations: CIMT, carotid intima media thickness; FMD, flow-mediated dilatation; MSNA, muscle sympathetic nerve activity; PWV, pulse wave velocity.

obese individuals with PCOS who had similar testosterone levels (Pan, 2023).

*Preclinical models.* Preclinical studies show that some, but not all, animal models of PCOS develop hypertension. Ovine models of PCOS (prenatal exposure to elevated testosterone) have blood pressure as adults that is ~10 mmHg higher than that of control animals (King et al., 2007), alongside most of the other metabolic characteristics of PCOS (Padmanabhan & Veiga-Lopez, 2013b). However data from rodent models are more equivocal, with the dihydrotestosterone (DHT)-treated models (Yanes et al., 2011), but not the dehydroepiandrosterone (DHEA)-treated (Oktanella et al., 2024) or testosterone propionate-treated (Utkan Korun et al., 2024) models developing elevated blood pressure *versus* controls. Interestingly, the DHEA-treated rodent model exhibited lower blood pressure *versus* control animals, which was unaffected by subsequent androgen receptor blockade (Perusquía et al., 2023). This highlights the heterogeneity of characteristics in rodent models of PCOS, in which metabolic, endocrine and ovarian features vary widely between models (Walters et al., 2012), whereas blood pressure appears to be elevated in individuals with PCOS and in large mammalian models. The following sections explore the mechanisms by which androgens could alter blood pressure regulation in PCOS.

**Androgens and sympathetic nerve activity in PCOS.** The sympathetic nervous system is a key determinant of blood pressure, modulating both vasoconstrictor tone and cardiac output (Dampney, 2016). Muscle sympathetic nerve activity (MSNA) correlates with resting blood pressure, both in hypertensive and normotensive individuals (Grassi et al., 2018), and chronically elevated MSNA is a hallmark of multiple cardiovascular diseases, including ischaemic heart disease and heart failure (Fisher et al., 2009).

*Clinical populations.* Direct measurements of sympathetic activity have revealed that MSNA is elevated in individuals with PCOS at rest (Lambert et al., 2015; Shorakae et al., 2018; Sverrisdóttir et al., 2008), with microneurographic recordings showing a 8–15 bursts/100 heartbeats or 6–10 bursts/min increase

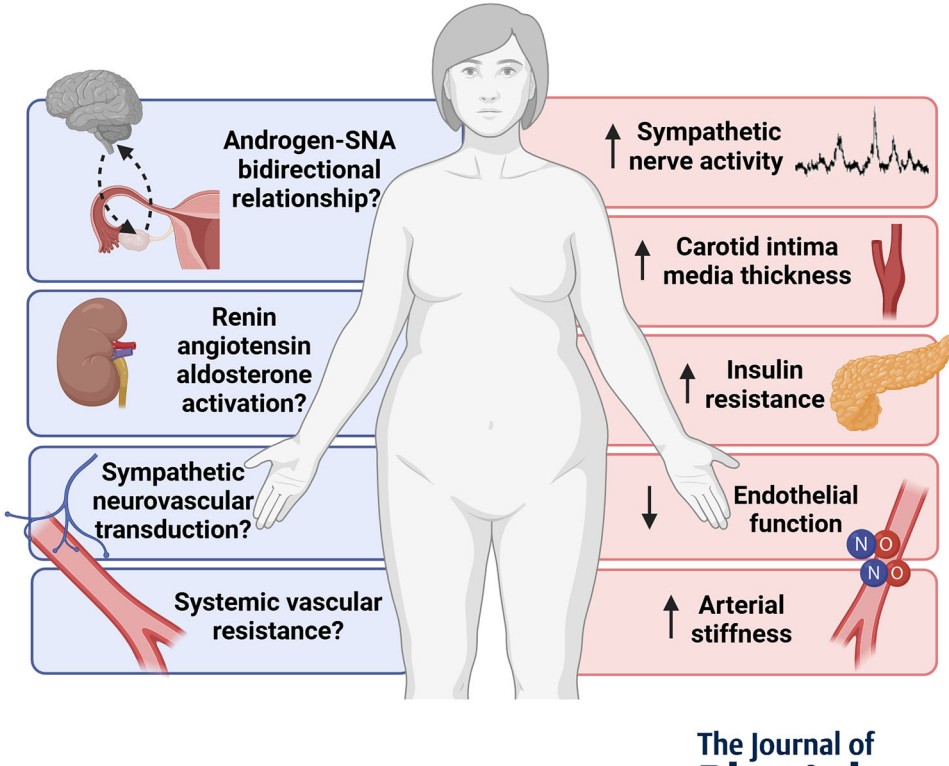

**Figure 1. The cardiovascular control mechanisms that may be altered in individuals with polycystic ovary syndrome (PCOS)**
Pink boxes indicate mechanisms for which there is evidence in individuals with PCOS; blue boxes indicate putative mechanisms with no current data available in human PCOS cohorts. SNA, sympathetic nerve activity; NO; nitric oxide.

**Table 2. Summary of the potential mechanisms by which androgens influence blood pressure regulation in several participant cohorts and preclinical models. These include both genomic and non-genomic actions of androgens.**

| Mechanism | Cohort/model | Reference |
|---|---|---|
| *Pro-hypertensive* | | |
| Increased muscle sympathetic nerve activity | Humans with androgen excess PCOS | (Lambert et al., 2015; Lansdown et al., 2019; Shorakae et al., 2018; Sverrisdóttir et al., 2008; Usselman, Coovardia et al., 2019) |
| Increased MC4R expression in brainstem | Hyperandrogenic female rat (DHT model) | (Maranon et al., 2015) |
| Androgen receptor-mediated activation of RVLM C1 neurones | Female rats without hyperandrogenism | (Milner et al., 2007) |
| Increased noradrenaline and neuropeptide Y production | Male rats | (Kumai et al., 1995; Sahu et al., 1989) |
| Increased $\alpha$-adrenergic receptor expression | Male spontaneously hypertensive rats | (McConnaughey & Iams, 1993) |
| Reduced eNOS expression and activity | Hyperandrogenic pregnant female rat | (Chinnathambi et al., 2013) |
| Reduced endothelium-derived hyperpolarising factor activity | Hyperandrogenic female rat (DHT model) | (Mishra et al., 2017) |
| Increased endothelin-1 expression | Humans with androgen excess PCOS | (Diamanti-Kandarakis et al., 2001; Orio et al., 2004; Wenner et al., 2013) |
| Impaired endothelin-1 B receptor-mediated vasodilatation | Humans with androgen excess PCOS | (Usselman, Yarovinsky et al., 2019; Wenner et al., 2013). |
| Upregulation of MMP-9 | Humans with androgen excess PCOS | (Lewandowski et al., 2006). |
| Downregulation of G protein-coupled oestrogen receptor | Hyperandrogenic female mouse (DHT model) | (Horton et al., 2024). |
| Increased expression of angiotensinogen, ACE and angiotensin II receptor type 1 | Hyperandrogenic female rat (DHT model) | (Yanes et al., 2011) |
| Reduced expression of angiotensin II receptor type 2 | Hyperandrogenic female rat (DHT model) | (Mishra et al., 2016) |
| *Anti-hypertensive* | | |
| Increased eNOS expression | Human endothelial cells | (Goglia et al., 2010; Yu et al., 2010) |
| Proliferation of endothelial cells | Human umbilical vein endothelial cells | (Converse & Thomas, 2021) |
| Activation of vascular $K^+$ channels | Human coronary artery endothelial cells | (Ruamyod et al., 2017) |
| Inactivation of vascular $Ca^{2+}$ channels | Human umbilical artery tissue | (Perusquía et al., 2007) |

Abbreviations: DHT, dihydrotestosterone; eNOS, endothelial nitric oxide synthase; MC4R, melanocortin-4 receptor; MMP-9, matrix metalloproteinase-9; PCOS, polycystic ovary syndrome; RVLM, rostral ventrolateral medulla.

in MSNA in individuals with PCOS *versus* controls. Additionally, indirect measures of sympathetic activity, such as plasma adrenaline concentrations, have been found to be elevated in PCOS compared to controls (Hashim et al., 2015). However, others have reported that resting MSNA was similar between individuals with PCOS and control groups, but that individuals with PCOS showed exaggerated MSNA responses to sympathoexcitatory stimuli (Lansdown et al., 2019; Usselman, Coovardia et al., 2019). Meanwhile, one study reported that MSNA was lower in individuals with PCOS *versus* controls (Stone et al., 2023). Notably MSNA has been positively associated with total and free testosterone in both normal weight (Sverrisdóttir et al., 2008) and overweight/obese individuals with PCOS (Shorakae et al., 2018). However this association is inconsistent, as Lambert and colleagues (2015) report no correlation between androgens and MSNA in overweight/obese

individuals with PCOS, despite observing elevated MSNA compared to controls (Lambert et al., 2015).

*Preclinical models.* Preclinical studies support the notion that increased blood pressure in PCOS is sympathetically mediated and have shed light on the possible underlying mechanisms. Although SNA has not been directly recorded in rodent models of PCOS, sympathetic inhibition (i.e. adrenergic blockade or renal denervation) prevents DHT-treated rats from developing the elevated blood pressure that normally occurs in this PCOS model (Maranon et al., 2015). Mechanistic data from the same study suggest that the melanocortin-4-receptor (MC4R), which activates sympathetic and inhibits parasympathetic preganglionic neurones in the brainstem (Girardet & Butler, 2014; Sohn et al., 2013), is important in modulating SNA and blood pressure in this PCOS model. That is, the expression of MC4R in the nucleus tractus solitarius was higher, and the depressor response to MC4R antagonism

was larger, in the hyperandrogenic rats *versus* controls (Maranon et al., 2015). Additionally MC4R-null rats did not develop DHT-induced increases in blood pressure (Maranon et al., 2015), suggesting that MC4R plays a key role in mediating the elevated blood pressure observed in this model of PCOS. In further support, clinical studies have shown that MC4R mediates the development of hypertension in overweight adults, as individuals with a loss-of-function mutation to MC4R had lower blood pressure, reduced hypertension prevalence and lower urinary noradrenaline metabolites than BMI- and insulin resistance-matched control participants (Greenfield et al., 2009). As such both preclinical evidence and clinical evidence indicate that MC4R may be important in mediating sympathetic activity in PCOS, although this has yet to be evaluated in individuals with PCOS.

Androgen receptors expressed in brain regions involved in baroreflex central processing could offer another mechanism by which androgens modulate sympathetic activity. That is, androgen receptors have been identified in the rostral ventrolateral medulla (RVLM) of adult male and female Sprague–Dawley rats on afferent fibres and glial cells of C1 adrenergic neurones (Milner et al., 2007). Importantly RVLM C1 adrenergic neurons project directly to sympathetic preganglionic neurons, and excessive activation has been directly linked to increased SNA (Dampney et al., 1985; Reis et al., 1988) and blood pressure (Reis et al., 1988; Wenker et al., 2017). Excess androgens, as seen in PCOS, may enhance binding to androgen receptors on RVLM C1 neurons, potentially exacerbating SNA. However, to the best of our knowledge, no studies have assessed the effects of testosterone applications to the brainstem on SNA, as has been investigated for oestrogen (Saleh et al., 2000). Additionally we did not find evidence that adrenergic receptor expression has been measured in the brainstem of the hyperandrogenic rodent models of PCOS. As such the modulation of SNA in PCOS by activation of brainstem androgen receptors remains a putative mechanism and requires further study.

**Androgens and sympathetic baroreflex control.** Sympathetic baroreflex sensitivity, the ability to buffer changes in blood pressure via modulation of MSNA, is a marker of baroreflex function and may be impaired in hypertensive individuals (Grassi et al., 2014).

*Clinical populations.* To date, only one study has assessed the sympathetic baroreflex in individuals with PCOS. Stone and colleagues (2023) showed that sympathetic baroreflex gain (forearm vascular resistance during baroreceptor unloading with lower body negative pressure) was similar in individuals with PCOS compared to controls. Despite this lack of group difference, manipulation of testosterone levels (suppression by

gonadotropin-release hormone (GnRH) antagonism, followed by re-introduction of exogenous testosterone) affected sympathetic baroreflex gain in PCOS but not in controls. The PCOS group had greater baroreflex gain under GnRH antagonism, which returned to control levels with the addition of exogenous testosterone, whereas the same procedures did not affect sympathetic baroreflex gain in controls (Stone et al., 2023). Therefore, although baseline sympathetic baroreflex gain appears to be similar in PCOS and control populations, these data suggest that testosterone may play a role in modulating the sympathetic baroreflex in PCOS that is absent in controls.

**Sympathetic-mediated androgen production.** Interestingly, the sympathetic nervous system may play a unique role in PCOS compared to other conditions with higher cardiovascular risk, as SNA may promote the production of androgens, which in turn increase SNA.

*Preclinical models.* Cold stress in rats increases noradrenaline activity, enhancing the activation of ovarian sympathetic nerves (Jara et al., 2010). This activation increases plasma testosterone levels and results in rats with PCOS-like phenotypes, including irregular oestrous cycles and thickened follicles that develop into cysts (Bernuci et al., 2008; Bernuci et al., 2013). When these cysts are reduced by acetylcholine administration, testosterone levels return to normal (Riquelme et al., 2021).

*Clinical populations.* In agreement, clinical data demonstrate higher levels of noradrenaline and dopamine in the follicles of PCOS patients compared to non-PCOS patients undergoing *in vitro* fertilisation (Musalı et al., 2016). As such these preclinical and clinical data highlight a potential bidirectional relationship between sympathetic activation and androgens.

In summary evidence suggests that MSNA is moderately elevated in PCOS (Hashim et al., 2015; Lambert et al., 2015; Lansdown et al., 2019; Shorakae et al., 2018; Sverrisdóttir et al., 2008), which may both drive and be driven by hyperandrogenism. Testosterone may play a role in modulating the sympathetic baroreflex in PCOS, although baroreflex gain is similar in PCOS and controls. The effect of sympathetic activity on blood pressure, however, ultimately depends on the conversion of MSNA into vasoconstrictor tone.

**Androgens and neurovascular transduction in PCOS.** Sympathetic neurovascular transduction describes the conversion of MSNA into vasoconstrictor tone. This process encompasses the arrival of sympathetic action potentials at the synaptic bulb, the release of neurotransmitters, the binding of these to vascular receptors and the resultant cell signalling cascades inside vascular

smooth muscle cells that initiate vasoconstriction (Young et al., 2021). In addition, the level of basal vasoconstrictor tone and any competing vasodilatory influences (e.g. nitric oxide (NO) levels, $\beta_2$-adrenergic receptor activity) also contribute to the vasoconstrictor response evoked by a given level of MSNA (Hart et al., 2011). As such changes in any of these processes could modulate the level of neurovascular transduction and thus alter the relationship between MSNA and blood pressure.

*Clinical populations.* Given the increased hypertensive risk in premenopausal individuals with PCOS (Amiri et al., 2020) and the elevated blood pressure in PCOS *versus* controls (Fu et al., 2023; Mellembakken et al., 2021), it is possible that sympathetic transduction is elevated in this cohort. At present, resting sympathetic transduction has not been directly measured in PCOS. However, Lansdown and colleagues (2019) showed that individuals with PCOS had greater sympathetic responses to an isometric hand grip stimulus compared to controls, whereas the pressor responses were similar between groups (Lansdown et al., 2019). Given that a larger increase in MSNA was needed to achieve the same increase in blood pressure, this may indicate *impaired* neurovascular transduction within PCOS. However, assessment using a signal averaging technique (for a review, see Young et al. (2021)) is needed to properly quantify sympathetic neurovascular transduction in PCOS, both at rest and during sympathoexcitatory stimuli.

## Androgens and vascular adrenergic signalling

*Preclinical models.* Preclinical work supports the notion that androgens impact various aspects of the sympathetic neurovascular transduction pathway, and thus could modulate transduction in PCOS, although there are currently little data from PCOS cohorts or preclinical models of PCOS. Several studies indicate that testosterone can modulate the production of the sympathetic neurotransmitters noradrenaline and neuropeptide Y (Kumai et al., 1995; Sahu et al., 1989). Specifically, castration of male spontaneously hypertensive rats reduced noradrenaline levels and adrenal expression of tyrosine hydroxylase, an effect that was reversed by testosterone replacement (Kumai et al., 1995). Similarly, testosterone replacement increased neuropeptide Y production in the hypothalamus of castrated male rats (Sahu et al., 1989). Meanwhile, other work has demonstrated that testosterone had no effect on electrically evoked noradrenaline release in male rodent mesenteric artery beds (Isidoro-Garcia et al., 2021). As such, testosterone appears to influence the production of sympathetic neurotransmitters but may not modulate neurotransmitter release at the synaptic bulb, at least in non-PCOS rodent models.

There is also evidence that androgens influence the expression and function of vascular adrenergic receptors. That is, testosterone exposure increased the expression of $\alpha$-1B adrenergic receptors in cultured hamster smooth muscle cells (sex not specified) (Sakaue & Hoffman, 1991) and restored normal levels of $\alpha$-adrenergic receptor expression in the tail arteries of castrated male rats (McConnaughey & Iams, 1993). In contrast, manipulation of testosterone levels by castration and testosterone replacement does not appear to alter vascular $\beta$-adrenergic receptor expression in male rats (Lopez-Canales et al., 2018; Riedel et al., 2019). However testosterone exposure at physiological levels (for male rats) reduced adenyl cyclase expression and levels of cyclic adenosine monophosphate in the rat aorta, indicating that testosterone could restrict $\beta$-adrenergic mediated vasodilatation, at least in males (Lopez-Canales et al., 2018). Overall, these preclinical data indicate that testosterone could enhance sympathetic neurovascular transduction via increases in sympathetic neurotransmitters and an upregulation of $\alpha$-adrenergic activity. However, we cannot identify any similar data from PCOS cohorts or models of chronic androgen exposure in females; therefore it remains to be established whether these mechanisms also occur in PCOS.

*Clinical populations.* Importantly, the balance of $\alpha$- and $\beta$-adrenergic receptor activity is thought to underlie the reduced sympathetic vascular transduction in healthy premenopausal females, given that systemic $\beta$-adrenergic blockade increases sympathetic vascular transduction (Briant et al., 2016; Hart et al., 2011). In this premenopausal female cohort, an oestrogen-related increase in basal NO production (Sudhir et al., 1996) is thought to increase $\beta_2$-adrenergic receptor sensitivity to noradrenaline (Kneale et al., 2000), which counteracts the vasoconstrictor effect of $\alpha$-adrenergic receptors (Hart et al., 2011). In PCOS, the balance between $\alpha$- and $\beta$-adrenergic activity could be shifted by the changes in receptor expression and activity described above; however this remains to be established.

## Androgens and vasoconstrictor tone

*Preclinical models.* There is preclinical evidence that androgens influence vasoconstrictor tone independent of sympathetic control. Many studies have examined the vasomotor response of resistance vessels *ex vivo* to acute testosterone exposure (see Lorigo et al. (2020)) for a comprehensive review). Although most studies have demonstrated a vasodilatory response to testosterone exposure (Crews & Khalil, 1999), some studies have demonstrated that testosterone facilitates the action of vasoconstrictors (Schrör et al., 1994). Acute vasodilatory responses to testosterone have also been observed *in vivo* in rodent (Perusquia et al., 2015) and porcine

models (Molinari et al., 2002). This acute vasodilatory response is observed even in the presence of aromatase inhibitors (Tep-areenan et al., 2002) and oestrogen receptor antagonists (Chou et al., 1996), indicating that vasodilatation is not mediated by the conversion of testosterone to oestrogen. The mechanisms underlying these acute vasodilatory responses to testosterone are discussed in detail elsewhere (Foradori et al., 2008; Lorigo et al., 2020; Lucas-Herald & Touyz, 2022; Lucas-Herald et al., 2017). In brief, these mechanisms appear to be both endothelium-dependent (Rowell et al., 2009), via an upregulation of NO (Yu et al., 2010), and endothelium-independent (Crews & Khalil, 1999; Tep-areenan et al., 2002), via the activation of $K^+$ channels (e.g. large and small conductance $Ca^{2+}$-activated $K^+$ channels ($BK_{Ca}$ and $SK_{Ca}$) (Ruamyod et al., 2017)) and the inactivation of voltage- and ligand-gated $Ca^{2+}$ channels (Perusquía et al., 2007) in the membrane of vascular smooth muscle cells.

*Clinical populations.* The vasodilatory actions of testosterone have been replicated in humans, both *ex vivo*, using donated blood vessels (Rowell et al., 2009; Seyrek et al., 2007), and *in vivo*, for example in the coronary arteries of older males (Webb et al., 1999). Importantly, these vasodilatory responses are still observed when male physiological concentrations of testosterone are used (Rowell et al., 2009). However, whether chronic exposure to testosterone (akin to the chronic hyperandrogenism in PCOS) alters basal vasoconstrictor tone is less clear. In a study of postmenopausal females using hormone replacement therapy, 6 weeks of testosterone exposure improved endothelium-dependent and endothelium-independent vasodilatation but did not alter resting brachial artery diameter (Worboys et al., 2001). To the best of our knowledge, only one study has assessed the vasomotor responses of resistance vessels from individuals with PCOS *ex vivo* (Kelly et al., 2002). Kelly and colleagues (2002) demonstrated that vasodilator and vasoconstrictor responses to acetylcholine and noradrenaline, respectively, did not differ between individuals with PCOS and control participants, suggesting that the ability of the vasculature to dilate and constrict is not impaired in PCOS. However, this does not provide insight into the resting vasoconstrictor tone in PCOS. Given the elevated MSNA and blood pressure in PCOS, it is plausible that individuals with PCOS operate at a higher resting vasoconstrictor tone, but this needs to be confirmed.

**Androgens and endothelial function in PCOS.** The vascular endothelium contributes to blood pressure by regulating vascular tone (Gallo et al., 2021), and endothelial dysfunction is characterised by an increase in vasoconstrictors acting on the endothelial cells (i.e.

endothelin-1 (ET-1)) (Deanfield et al., 2005) and a reduction in the bioavailability of vasodilators (i.e. NO) (Brunner et al., 2005). The non-invasive flow-mediated dilatation (FMD) technique is commonly used to assess endothelial dysfunction in conduit vessels, correlates negatively with blood pressure in normotension and hypertension (Maruhashi et al., 2013) and predicts future cardiovascular events (Celermajer, 1997). Notably, there is compelling evidence for endothelial dysfunction, as assessed by FMD, in both lean (Berbrier et al., 2023; Cussons et al., 2009; Orio et al., 2004; Sorensen et al., 2006; Soyman et al., 2011; Tarkun et al., 2004) and overweight/obese individuals with PCOS (Cascella et al., 2008; Diamanti-Kandarakis et al., 2006; Kravariti et al., 2005; Meyer et al., 2005; Sprung et al., 2014) compared to controls. Although this is not a universal finding (Krentowska et al., 2021), a meta-analysis concluded that there was a 3.4% impairment in FMD in individuals with PCOS *versus* controls (Sprung et al., 2013). This likely represents a clinically meaningful change, given that a 1% reduction in FMD represents an 8%–12% increase in cardiovascular risk in a combined clinical and non-clinical population (Ras et al., 2013). One study demonstrated that endothelial function was negatively correlated with androgen levels in lean (but not in overweight/obese) individuals with PCOS (Berbrier et al., 2023), which suggests that hyperandrogenism may underlie endothelial dysfunction in lean individuals with PCOS.

*Preclinical models.* It has been suggested that hyperandrogenism impairs endothelial function by reducing or limiting NO production (Stanhewicz et al., 2018). In support of this, in a hyperandrogenic (pregnant) rat model, NO-mediated dilatation in mesenteric (Chinnathambi et al., 2013) and uterine arteries (Chinnathambi et al., 2014) was reduced. Moreover, elevated testosterone was associated with reduced plasma levels of NO metabolites, reduced endothelial NO synthase (eNOS) protein expression in the uterine arteries (Chinnathambi et al., 2014) and decreased eNOS activity in the mesenteric arteries (Chinnathambi et al., 2013). Furthermore, although arteries from the DHT rat model of PCOS showed impaired endothelium-dependent vasodilatation, this was reversed with exposure to the androgen receptor antagonist flutamide (Hurliman et al., 2015). As such, there is evidence for an androgen-mediated downregulation of NO production in PCOS. However this must be reconciled with the large evidence base suggesting that testosterone is vasodilatory (Lorigo et al., 2020; Lucas-Herald & Touyz, 2022), activates eNOS in human endothelium (Goglia et al., 2010; Yu et al., 2010) and increases human vascular endothelial cell proliferation (Converse & Thomas, 2021). This discrepancy could arise from differences in the durations of exposure to androgens

(i.e. acute application to cells/tissues *versus* chronic hyperandrogenism in PCOS). However, chronic testosterone supplementation in other female cohorts improved FMD responses (Worboys et al., 2001). Therefore it is worth considering alternative mechanisms that could be responsible for the impairment in FMD observed in individuals with PCOS.

*Clinical populations.* Given that two studies showed normal endothelium-dependent vasodilatation evoked by non-shear stimuli (acetylcholine/bradykinin) in PCOS (Kelly et al., 2002; Ketel et al., 2008), it is possible that impaired FMD is related to the shear stimulus. The only study to report shear rate found a borderline significant ($p < 0.06$) difference in this metric between PCOS and controls (Berbrier et al., 2023), but given the large impairment in FMD seen in PCOS, it is unlikely that this is fully explained by differences in shear, and it is unclear how androgens might impact the shear stimulus. Alternatively, endothelial vasodilators other than NO (such as endothelium-derived hyperpolarising factor (EDHF)) may be downregulated, given that in the DHT rat model of PCOS, endothelium-dependent vasodilatation was still reduced *versus* controls when the vasodilatory effect of EDHF was isolated from that of NO and prostaglandins (Mishra et al., 2017). On the contrary, the ability of the vascular smooth muscle to relax might be impaired. Indeed, two studies support a reduced endothelium-independent vasodilatation in individuals with PCOS *versus* controls (Dokras et al., 2006; Kravariti et al., 2005), although another study reported no difference (Sorensen et al., 2006). However, it is unclear how hyperandrogenism may impair vascular smooth muscle relaxation in PCOS, given that testosterone has a direct vasorelaxant effect via several mechanisms (for a review, see Lorigo et al. (2020)). Another possibility is that vasoconstricting factors limit the vasodilatory response to the FMD stimulus. Notably, individuals with androgen excess PCOS exhibit increased serum levels of ET-1 (Diamanti-Kandarakis et al., 2001; Diamanti-Kandarakis et al., 2006; Orio et al., 2004) compared to BMI-matched controls. ET-1 acts as a vasoconstrictor at vascular smooth muscle via the $ET_A$ and $ET_B$ receptors but also has vasodilatory actions when binding $ET_B$ receptors at the vascular endothelium (Mazzuca & Khalil, 2012). In PCOS the vasodilatory action of ET-1 is impaired (Usselman, Yarovinsky et al., 2019; Wenner et al., 2013). Therefore, both increased ET-1 levels and impaired $ET_B$ receptor-mediated vasodilatation appear to contribute to a vasoconstrictor effect of ET-1 in PCOS. This, together with the increased MSNA in PCOS, may act to limit the vasodilator capacity and therefore the FMD response in PCOS.

**Insulin resistance and endothelial function.** It is important to note that characteristics of PCOS other than hyperandrogenism may contribute to the impaired FMD response observed. For example, a negative correlation between insulin resistance and endothelial function has been observed in individuals with PCOS (Kravariti et al., 2005; Paradisi et al., 2001). The vasodilatory effects of insulin are well established (Baron, 1994) and involve the activation of eNOS via the PI3K/Akt signalling pathway following endothelial insulin receptor activation (Kim et al., 2006). In insulin resistance, it is proposed that the PI3K/Akt pathways are downregulated, reducing NO production (Kahn, 1985), whereas the mitogen-activated protein kinase insulin signal pathways, which regulate the secretion of ET-1, remain intact (Kim et al., 2006). Consequently, this leads to a reduction in available NO and a relative increase in ET-1, both of which contribute to endothelial dysfunction. Lastly, although insulin plays a critical role in endothelial function, its effects are often compounded by elevated androgens. Increased insulin level has been shown to mimic the actions of luteinizing hormone (i.e. stimulates ovarian theca cells to synthesise androstenedione and testosterone; Bienenfeld et al. (2019)), leading to increased ovarian androgen secretion (Nestler et al., 1998; Wu et al., 2014). Additionally the overactivity of the protein kinase B/Akt signalling pathways (associated with insulin resistance; Li et al. (2017)) may lead to increased theca cell stimulation and further increases androgen secretion in PCOS (Ye et al., 2021). This overproduction of androgens, in combination with insulin resistance, exacerbates the prevalence of endothelial dysfunction in individuals with PCOS, highlighting the complex interplay between metabolic and hormonal disturbances in this population.

**Androgens and the renin-angiotensin-aldosterone system in PCOS.** Given the role of the renin-angiotensin-aldosterone system (RAAS) in elevating blood pressure, it is possible that this system also contributes to the increased hypertension prevalence in PCOS. Data on whether RAAS is elevated in PCOS compared to controls are contradictory, with some studies reporting elevated plasma renin (Alphan et al., 2013; Hacihanefioglu et al., 2000; Jaatinen et al., 1995), angiotensin II (Arefi et al., 2013) or aldosterone (Cascella et al., 2006) *versus* controls, whereas others report no difference (Stone et al., 2023). Meanwhile, some studies report differences in some markers of RAAS activity but not others (e.g. elevated renin but not angiotensin II) (Alphan et al., 2013; Cascella et al., 2006). Whether any change in RAAS in PCOS is mediated by hyperandrogenism is also unclear. Although some studies report correlations between androgens and markers of RAAS (Jaatinen et al., 1995), others report no relationship

(Cascella et al., 2006). The effects of sex hormones on RAAS have been reviewed in detail previously (Medina et al., 2020; te Riet et al., 2015; White et al., 2019). Current evidence broadly suggests that oestrogen tends to promote the vasodilatory arm of the RAAS (mediated by angiotensin-converting enzyme (ACE)-2, angiotensin II receptor type 2 and Mas receptor) more than the vaso-constrictor arm (mediated by ACE and angiotensin II receptor type 1 ($AT_1R$)), whereas testosterone appears to do the opposite (Medina et al., 2020; te Riet et al., 2015). However, much of this evidence does not come from individuals with PCOS or preclinical models of PCOS. As such, the overall effect of sex hormones on RAAS in PCOS is difficult to assess.

*Preclinical models.* Several studies have tested whether androgens alter RAAS activity in preclinical models of PCOS. Female rats treated with DHT showed increased expression of angiotensinogen, ACE and angiotensin II receptor type 1 (Yanes et al., 2011) but reduced expression of angiotensin II receptor type 2 (Mishra et al., 2016), suggesting that testosterone excess in female animal models promotes the vasoconstrictor over the vaso-dilatory arm of RAAS. Additionally RAAS inhibition (ACE inhibitors) reduced blood pressure in the DHT rat model of PCOS (Torres Fernandez et al., 2018). Meanwhile, androgen receptor blockade by flutamide reduced blood pressure and plasma renin, compared to untreated controls, in a female transgenic rat model with high RAAS activity (Baltatu et al., 2003). As such, pre-clinical evidence supports the possibility that androgens could promote hypertension via RAAS activation.

*Clinical populations.* Experimental data on RAAS in individuals with hyperandrogenic PCOS are scarce. Stone and colleagues (2023) reported that the block of sex hormone production (GnRH antagonism) reduced aldosterone concentrations in PCOS but not in age-, BMI- and insulin-resistance-matched controls (Stone et al., 2023), suggesting a role of androgens in RAAS activation in PCOS. In contrast, another study also using GnRH antagonism showed that individuals with PCOS produced more aldosterone in response to exercise *versus* controls, even under blockade of sex hormone production (Stachenfeld et al., 2010), indicating that the responsiveness of RAAS was still elevated in PCOS when androgen production was inhibited. This discrepancy may be explained by the chronic effects of hyperandrogenism on RAAS (i.e. receptor expression) that may not have been removed by temporary GnRH treatment. However these limited studies, together with the preclinical evidence, suggest that the interaction between RAAS and androgens requires further study in PCOS. Of note, RAAS activation promotes some of the pro-hypertensive mechanisms discussed elsewhere in this review (e.g. vasoconstriction via $AT_1R$, increased sympathetic nerve activity) (Miller &

Arnold, 2019) and could therefore contribute to elevated blood pressure in PCOS via several pathways.

**Androgens and vascular structure in PCOS.** Finally, PCOS has been associated with changes to arterial structure that could contribute to elevated blood pressure. Arterial stiffness is associated with cardiovascular disease risk (Laurent et al., 2006; Van Bortel et al., 2012; Vlachopoulos et al., 2010) and is thought to occur before the development of hypertension (Oh et al., 2017).

*Clinical populations.* Studies in lean (Abacioglu et al., 2021; Kilic et al., 2021; Sasaki et al., 2011; Soares et al., 2009) and overweight/obese (Kelly et al., 2002; Meyer et al., 2005) individuals with PCOS report that stiffness in various arteries (as measured by pulse wave velocity (PWV)) is elevated by between 0.36 and 0.81 m/s compared to BMI-matched controls. However several studies have also reported no change in arterial stiffness in PCOS (Burlá et al., 2019; Cussons et al., 2009; Ketel et al., 2010; Kim et al., 2019; Rees et al., 2014). Meanwhile, some echocardiographic studies report increased aortic stiffness and reduced aortic distensibility in individuals with PCOS *versus* controls (Gencer et al., 2014; Lakhani et al., 2006), whereas another study reports no difference (Kaya et al., 2010). As such, there are conflicting findings regarding arterial stiffness in PCOS. However, given that a 1 m/s increase in aortic PWV corresponds to a 15% increase in all-cause mortality in the general population (Vlachopoulos et al., 2010), an increase in aortic stiffness in PCOS of the magnitude reported by some studies could be clinically relevant. Furthermore, aortic stiffness is an independent predictor of hypertension in normotensive adults (Dernellis & Panaretou, 2005), and as such could contribute to the increased prevalence of hypertension in PCOS.

*Preclinical models.* Arterial stiffness is correlated with androgens in PCOS (Burlá et al., 2019; Kilic et al., 2021; Soares et al., 2009), and it is possible that androgens alter the cellular and extracellular composition of arteries. That is, increased androgen levels induced by administration of DHT in female mice were correlated with decreases in collagen (Horton et al., 2024), and the elastin/collagen ratio of human aortic smooth muscle cells (indicative of reduced vascular distensibility) exposed to testosterone was found to be 11 times lower than when exposed to oestradiol and progesterone (Natoli et al., 2005). Moreover, serum concentrations of matrix metalloproteinase-9 (MMP-9), a proteolytic enzyme that can degrade elastic components of the arterial wall and promotes arterial stiffening (Yasmin et al., 2005), were elevated in individuals with PCOS (Lewandowski et al., 2006). Additionally, DHT was shown to downregulate expression of G protein-coupled oestrogen receptor (GPER) (Horton et al., 2024), which has protective effects

against vascular stiffening, given that GPER knockout increased arterial stiffness in female mice (Ogola et al., 2021). As such, MMP-9 activity and GPER expression are potential mechanisms by which androgens could alter arterial stiffness in PCOS.

**Carotid intima media thickening and PCOS.** Other structural changes to arteries, such as increased carotid intima media thickness (CIMT), could also influence blood pressure in PCOS, given that blood pressure correlates positively with CIMT (Ferreira et al., 2016). A meta-analysis reported an increase of 0.072 mm in individuals with PCOS *versus* controls, which is comparable to 7 years of age-related increase in CIMT (Meyer et al., 2012). Intima media thickening at the carotid bulb has been linked to reduced cardiovagal baroreflex sensitivity (Gianaros et al., 2002), possibly due to changes in arterial distension and baroreceptor loading (Bonyhay et al., 1996; Lage et al., 1993). Therefore it is possible that increased CIMT influences blood pressure regulation in PCOS. In addition, CIMT was positively correlated with levels of several androgens in PCOS (Luque-Ramírez et al., 2007), suggesting that androgens may contribute to elevated CIMT in PCOS.

## Conclusions

Elevated blood pressure is a key risk factor driving excess cardiovascular risk in PCOS. Case-control studies now show that individuals with PCOS exhibit some of the same changes in blood pressure regulation that occur in other cohorts at risk of hypertension, including endothelial dysfunction, sympathetic nervous system activation, RAAS activation and arterial stiffening. Meanwhile, other aspects of blood pressure control have not yet been adequately studied in PCOS (sympathetic neurovascular transduction and vasoconstrictor tone). The vascular actions of testosterone have been well studied in preclinical experiments, and this body of work is often drawn upon to explain the findings of clinical studies in individuals with PCOS. Sometimes preclinical studies support the clinical evidence, for example in the over-activation of the sympathetic nervous system in PCOS (Maranon et al., 2015; Sverrisdóttir et al., 2008). Yet in other cases preclinical work profoundly disagrees with the observations from clinical studies, for example the vasodilatory and NO-promoting effects of testosterone in cellular and tissue studies (Yu et al., 2010) but the observed impairment in endothelial function in individuals with PCOS (Sprung et al., 2013). These discrepancies highlight the need for a more nuanced understanding of the role of androgens in PCOS. In particular we believe there is a need for more detailed consideration of whether pre-clinical work is modelling hyperandrogenism in physio-logically relevant terms for PCOS (i.e. a chronic, female exposure to androgens at the physiological concentrations that occur in PCOS (not at male levels)). Furthermore, as there is now evidence that high testosterone has different effects on metabolic disease risk in males and females (Ruth et al., 2020), researchers may need to be cautious when using data or samples from male participants to understand the actions of androgens on the vasculature in PCOS. As mechanistic (rather than observational) human studies of PCOS begin to emerge (Berbrier et al., 2023; Stone et al., 2023), a better understanding of the preclinical evidence on androgens and blood pressure will help to inform the design of clinical work. Given the early age at which individuals with PCOS are exposed to increased cardiovascular risk, understanding the pathophysiological mechanisms in this cohort is crucial in identifying strategies to mitigate this risk and improving outcomes for such a considerable proportion of the female population.

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

# Additional information

## Competing interests

The authors declare no competing interests.

## Author contributions

All authors were responsible for concept and design of the work. Z.A., D.B., B.S., W.H. and C.R. drafted the manuscript. All authors revised the manuscript critically for important intellectual content. All authors approve the final version and agree to be accountable for all aspects of the work in ensuring that questions related to the accuracy or integrity of any part of the work are appropriately investigated and resolved. All persons designated as authors qualify for authorship, and all those who qualify for authorship are listed.

## Funding

Z.A. is supported by an Academy of Medical Sciences Springboard Grant (SBF007\100 185, PI R.L.). D.B. is supported by Fonds de recherche du Québec – Santé (FRQS). C.R. is supported by The Waterloo Foundation (grant no. 515 818). W.H is supported by Canadian Institute of Health Research (CIHR).

## Keywords

androgen, blood pressure, female, neurovascular, polycystic ovary syndrome, sympathetic

## Supporting information

Additional supporting information can be found online in the Supporting Information section at the end of the HTML view of the article. Supporting information files available:

**Peer Review History**

