## [Peer Review History · The Journal of Physiology]

The impact of androgens on cardiovascular control mechanisms in polycystic ovary syndrome: recent advances and translational approaches.

Zoe H. Adams, Danielle E. Berbrier, Brittany K. Schwende, Will Huckins, Cory T Richards, Aled Rees, Charlotte W. Usselman, and Rachel N Lord
DOI: 10.1113/JP287288

Corresponding author(s): Zoe Adams (ZAdams@cardiffmet.ac.uk)

The following individual(s) involved in review of this submission have agreed to reveal their identity: Matthew Babcock (Referee #1)

Review Timeline:	Submission Date:	19-Nov-2024
	Editorial Decision:	08-Jan-2025
	Revision Received:	14-Mar-2025
	Accepted:	24-Mar-2025

Senior Editor: Laura Bennet

Reviewing Editor: Christopher Lear

Transaction Report:

Dear Dr Adams,

Re: JP-TR-2024-287288 "The impact of androgens on neurovascular blood pressure control in polycystic ovary syndrome: recent advances and translational approaches." by Zoe H. Adams, Danielle E. Berbrier, Brittany K. Schwende, Will Huckins, Cory Richards, Aled Rees, Charlotte W. Usselman, and Rachel N Lord

Thank you for submitting your manuscript to The Journal of Physiology. It has been assessed by a Reviewing Editor and by 2 expert referees and we are pleased to tell you that it is acceptable for publication following satisfactory revision.

ABSTRACT FIGURES: Authors may use The Journal's premium BioRender account to create/redraw their Abstract Figures (and any other suitable schematic figure). Information on how to access this account is here: <https://physoc.onlinelibrary.wiley.com/journal/14697793/biorender-access>.

REVISION CHECKLIST: Upload a full Response to Referees file. To create your 'Response to Referees' copy all the reports, including any comments from the Senior and Reviewing Editors, into a Microsoft Word, or similar, file and respond to each point, using font or background colour to distinguish comments and responses and upload as the required file type.

We look forward to receiving your revised submission.

Yours sincerely,

Laura Bennet

EDITOR COMMENTS

Reviewing Editor:

Thank you for submitting your well written and synthesised review to the Journal of Physiology. The reviewers have been positive about the importance of this review to the field. Please address their suggestions for improvement.

Please also see 'Required Items' below.

REFEREE COMMENTS

Referee #1:

In this manuscript, Dr. Adams and colleagues review the effects of androgens on neurovascular blood pressure control in human and animal models of PCOS. This well-written review is especially useful because it highlights inconsistencies with the available pre-clinical models of PCOS and contrasts findings from those models with data from humans. Considering the complexity of PCOS, the authors have done a commendable job of synthesizing the available data. There are several weaknesses that should be addressed.

Line 120-121: It would be helpful to define what you mean by "direct and indirect measurements of sympathetic activity". My assumption was that this was referring to direct recordings of sympathetic activity vs. norepinephrine concentrations, but then the subsequent sentences appear to only refer to recordings of MSNA.

In some sections (Androgens and sympathetic nerve activity in PCOS, for example), the text jumps back and forth between human and animal data. It might be helpful to the reader to separate these sections, even with just a new paragraph. Line 133 is an example where a very long paragraph containing both human and animal data could be split up.

It may be helpful to include data from males in the baroreflex section (Lines 166-176), similar to their inclusion in the "Androgens and vascular adrenergic signalling" section (Lines 215-250). There are a few papers that I am aware of in male rodents demonstrating reduced BRS with testosterone suppression that is restored via testosterone add-back (PMIDs: 15799780, 16430770, 11602822). As these are in males and therefore may not be appropriate comparisons to individuals with PCOS, their inclusion may not be appropriate.

One aspect of blood pressure control that is seemingly overlooked in this review is the renin-angiotensin-aldosterone system. Considering the interactions of the RAAS system with both autonomic control of blood pressure and vascular function, it is worth discussing the available evidence, even if only briefly. PMIDs: 16940454, 20195177, 37327000, 1568764 (male rats), 2723066 (male rats), 7720907, and 21536229).

Referee #2:

The manuscript by Adams and colleagues aims to provide a comprehensive review of the factors contributing to higher blood pressure (and altered BP regulation) in women with PCOS. This is a critically important area for women's health, given this is the most common endocrine disorder for women and has clinical implications for CVD risk. The review paper is well-written and provides a well-rounded summary of the current literature. Importantly, the manuscript also has an important call to action for the scientific community to further address several unanswered questions about PCOS. Below are comments

for the authors to consider.

Title: Consider changing the title to include the broad information presented in the review. The paper is very well-written and covers several different aspects of cardiovascular function. The title and purpose have Neurovascular BP Control as the focus - however there is a big portion of the review that is focused on other vascular signaling mechanisms and endothelial function. While they may all be integrated, I wouldn't want some readers to skip over this excellent review if the title is too narrowly focused.

Along the same lines, could Fig 1 be updated? The authors spend time throughout the review talking about lean vs obese PCOS, along with other vascular signaling mechanisms and the role of insulin. Could that somehow be added to Fig 1 as well?

Fig 1- MSNA is shown in the brain / head - but this is measured in the periphery (muscle sympathetic nerve activity. It also is not correct to indicate MSNA to the ovary (not "muscle" sympathetic nerve activity). Consider reframing the location of 'MSNA' or perhaps just referring to SNA?

The authors may wish to consider adding a paragraph or small section on the Renin-Angiotensin-Aldosterone system as this is a primary regulator of BP and is overactive in PCOS.

Baroreflex paragraph (lines 166-176) - Can the authors clarify gain vs sensitivity? In one sentence it states that sensitivity is different but gain unaltered? Also, the last sentence of this paragraph states testosterone influences sympathetic baroreflex but sensitivity is not altered. Which aspects of the baroreflex curve then are impacted by testosterone?

In the conclusions (line 428) - The authors state that using male T concentrations or male models to understand hyperandrogenism or how that is used as a model of PCOS - This seems like a very important point to make earlier in the review when discussing the differences between preclinical or cell models and humans. The authors may wish to make this more clear earlier in the review.

Table 1 - while the table is a nice summary of some of the papers presented, the authors may wish to consider adding additional summary figures or key figures from some of the papers presented.

REQUIRED ITEMS

- Please include an Abstract Figure file, as well as the Figure Legend text within the main article file. The Abstract Figure is a piece of artwork designed to give readers an immediate understanding of the Review Article and should summarise the main conclusions. If possible, the image should be easily 'readable' from left to right or top to bottom. It should show the physiological relevance of the Review so readers can assess the importance and content of the article. Abstract Figures should not merely recapitulate other figures in the Review. Please try to keep the diagram as simple as possible and without superfluous information that may distract from the main conclusion of the Review. Abstract Figures must be provided by authors no later than the revised manuscript stage and should be uploaded as a separate file during online submission labelled as File Type 'Abstract Figure'. Please ensure that you include the figure legend in the main article file. All Abstract Figures will be sent to a professional illustrator for redrawing and you may be asked to approve the redrawn figure before your paper is accepted.

- Please upload separate high quality figure files via the submission form.

- Author profile(s) must be uploaded via the submission form. Authors should submit a short biography (no more than 100 words for one author or 150 words in total for two authors) and a portrait photograph of the two leading authors on the paper. These should be uploaded and clearly labelled together in a Word document with the revised version of the manuscript. Any standard image format for the photograph is acceptable, but the resolution should be at least 300 DPI and preferably more. A group photograph of all authors is also acceptable, providing the biography for the whole group does not exceed 150 words.

- Please ensure that the Article File you upload is a Word file.

END OF COMMENTS

Dr Zoe Adams
Cardiff School of Sport and Health Sciences
Cardiff Metropolitan University

Dr Laura Bennet
Senior Editor
The Journal of Physiology

4th March 2025

Manuscript ID: JP-TR-2024-287288

Revised Title: The impact of androgens on cardiovascular control mechanisms in polycystic ovary syndrome: recent advances and translational approaches.

Corresponding Author: Dr. Zoe Adams

Dear Dr. Bennet,

Thank you for requesting revisions to our manuscript and allowing us to resubmit for consideration of publication in *Journal of Physiology*. We appreciate the positive comments of the reviewers which highlighted the manuscript's synthesis of pre-clinical and human evidence in polycystic ovary syndrome (PCOS) and its emphasis on addressing the key gaps in PCOS research. We appreciate the thoughtful comments of the reviewers and have incorporated their suggestions, which we believe have strengthened the manuscript considerably.

Please see attached for clean and redlined versions of the manuscript and figures. Below we have provided detailed responses to individual reviewer comments (our responses in **bold**).

Reviewer #1:

In this manuscript, Dr. Adams and colleagues review the effects of androgens on neurovascular blood pressure control in human and animal models of PCOS. This well-written review is especially useful because it highlights inconsistencies with the available pre-clinical models of PCOS and contrasts findings from those models with data from humans. Considering the complexity of PCOS, the authors have done a commendable job of synthesizing the available data. There are several weaknesses that should be addressed.

Response: Thank you for your positive comments. We appreciate the time taken to review the manuscript. We believe that your thoughtful comments have improved the review.

Comment 1: Line 120-121: It would be helpful to define what you mean by "direct and indirect measurements of sympathetic activity". My assumption was that this was referring to direct recordings of sympathetic activity vs. norepinephrine concentrations, but then the subsequent sentences appear to only refer to recordings of MSNA

Response: Thank you for pointing this out. This section has been revised to clarify the distinction between direct and indirect measures of sympathetic activity in PCOS (Page 7, lines 126-131).

Comment 2: In some sections (Androgens and sympathetic nerve activity in PCOS, for example), the text jumps back and forth between human and animal data. It might be helpful to the reader to separate these sections, even with just a new paragraph. Line 133 is an example where a very long paragraph containing both human and animal data could be split up.

Response: Thank you for this comment. We have revised the manuscript to improve clarity by adding subheadings throughout titled 'Preclinical models' and 'Clinical populations' and structuring the text into separate paragraphs to distinguish between human and animal studies. This has meant some edits to the 'androgens and vasoconstrictor tone' section (pages 14-15, lines 276-296). In some cases, we have still included human and animal data in the same paragraph, where the same mechanism is being discussed (e.g., MC4R), as we felt it was important to highlight to readers where preclinical and clinical evidence converge.

Comment 3: It may be helpful to include data from males in the baroreflex section (Lines 166-176), similar to their inclusion in the "Androgens and vascular adrenergic signalling" section (Lines 215-250). There are a few papers that I am aware of in male rodents demonstrating reduced BRS with testosterone suppression that is restored via testosterone add-back (PMIDs: 15799780, 16430770, 11602822). As these are in males and therefore may not be appropriate comparisons to individuals with PCOS, their inclusion may not be appropriate

Response: Thank you for your comment and for suggesting these papers. We agree that data in male rodent models can provide valuable insight into baroreflex sensitivity and the potential implications of testosterone. However, to the best of our knowledge, we were unable to identify any studies—either in male rodent models or human participants—that specifically examine sympathetic baroreflex sensitivity and its modulation, which is the primary focus of this section in our review. Given this, we have opted not to include these studies, as they may not directly align with our discussion.

Comment 4: One aspect of blood pressure control that is seemingly overlooked in this review is the renin-angiotensin-aldosterone system. Considering the interactions of the RAAS system with both autonomic control of blood pressure and vascular function, it is worth discussing the available evidence, even if only briefly. PMIDs: 16940454, 20195177, 37327000, 1568764 (male rats), 2723066 (male rats), 7720907, and 21536229).

Response: Thank you for pointing this out. We have included a new section briefly discussing the evidence for a role of RAAS in promoting hypertension in individuals with PCOS and preclinical models of PCOS (pages 19-21, lines 393-436). The interactions

between sex hormones and RAAS are complex and beyond the scope of our review, so we have referred to some detailed reviews in this area for interested readers.

Reviewer #2:

The manuscript by Adams and colleagues aims to provide a comprehensive review of the factors contributing to higher blood pressure (and altered BP regulation) in women with PCOS. This is a critically important area for women's health, given this is the most common endocrine disorder for women and has clinical implications for CVD risk. The review paper is well-written and provides a well-rounded summary of the current literature. Importantly, the manuscript also has an important call to action for the scientific community to further address several unanswered questions about PCOS. Below are comments for the authors to consider.

Response: Thank you for your positive comments. We appreciate your comments and the time taken to review the manuscript.

Comment 1: Title: Consider changing the title to include the broad information presented in the review. The paper is very well-written and covers several different aspects of cardiovascular function. The title and purpose have Neurovascular BP Control as the focus - however there is a big portion of the review that is focused on other vascular signaling mechanisms and endothelial function. While they may all be integrated, I wouldn't want some readers to skip over this excellent review if the title is too narrowly focused.

Response: Thank you for your comment; this is an excellent point. We have updated the title and have revised the purpose accordingly to better reflect the broad scope of this review (pages 1-2, 4, lines: 14, 58-62).

Comment 2: Along the same lines, could Fig 1 be updated? The authors spend time throughout the review talking about lean vs obese PCOS, along with other vascular signaling mechanisms and the role of insulin. Could that somehow be added to Fig 1 as well?

Response: Thank you for this comment, we agree it is important to convey the role of insulin resistance and obesity in PCOS. We have revised Figure 1 so that it also refers to insulin resistance (and RAAS). We have included a new Table (1) that discusses how insulin resistance and obesity modulate the increased cardiovascular risk in PCOS. We felt it was clearer to present this information as a table and using a table allows us to include references. The original Table 1 has become table 2.

Comment 3: Fig 1- MSNA is shown in the brain / head - but this is measured in the periphery (muscle sympathetic nerve activity. It also is not correct to indicate MSNA to the ovary (not "muscle" sympathetic nerve activity). Consider reframing the location of 'MSNA' or perhaps just referring to SNA?

Response: Thank you for pointing out this error. We have revised Figure 1 to indicate a relationship between ‘SNA’ and androgen production.

Comment 4: The authors may wish to consider adding a paragraph or small section on the Renin-Angiotensin-Aldosterone system as this is a primary regulator of BP and is overactive in PCOS.

Response: Thank you for your comment which was shared with Reviewer #1. We have added a short section discussing the renin-angiotensin-aldosterone system, given its significant role in blood pressure regulation (pages 19-21, lines 393-436).

Comment 5: Baroreflex paragraph (lines 166-176) - Can the authors clarify gain vs sensitivity? In one sentence it states that sensitivity is different but gain unaltered? Also, the last sentence of this paragraph states testosterone influences sympathetic baroreflex but sensitivity is not altered. Which aspects of the baroreflex curve then are impacted by testosterone?

Response: Thank you for pointing out this error. We have revised the paragraph (page 10, lines 181-190) and hope that it now accurately reflects the findings from Stone and colleagues, who assessed sympathetic baroreflex *gain*. They found that whilst there was no difference in gain between PCOS and controls at baseline, the manipulation of testosterone levels by GnRH antagonism and reintroduction of exogenous testosterone altered baroreflex gain in PCOS but not controls (gain increased with GnRH antagonism in PCOS, no change in controls; gain reduced back to baseline levels with GnRH antagonism + testosterone in PCOS, no change in controls). Whilst these data may appear contradictory, we believe they indicate a potential role for testosterone in influencing the sympathetic baroreflex in PCOS that is absent in controls, even when baroreflex gain in PCOS is similar to that of controls.

Comment 6: In the conclusions (line 428) - The authors state that using male T concentrations or male models to understand hyperandrogenism or how that is used as a model of PCOS - This seems like a very important point to make earlier in the review when discussing the differences between preclinical or cell models and humans. The authors may wish to make this more clear earlier in the review.

Response: Thank you for your comment, this is a great point. We have added this clarification earlier in the review when discussing differences between preclinical models and human studies (page 4, lines 58-62).

Comment 7: Table 1 - while the table is a nice summary of some of the papers presented, the authors may wish to consider adding additional summary figures or key figures from some of the papers presented.

Response: Thank you for your comment, we have made some changes to the figures and tables. We have added to the original Table 1 (now called Table 2), now that additional

studies have been included (e.g., the RAAS section). We have added a new table, which summarises the cardiovascular risk factors present in PCOS and the effect of obesity and insulin resistance on these. Figure 1 now summarises all the mechanisms discussed in the review.

Dear Dr Adams,

Re: JP-TR-2025-287288R1 "**The impact of androgens on cardiovascular control mechanisms in polycystic ovary syndrome: recent advances and translational approaches.**" by Zoe H. Adams, Danielle E. Berbrier, Brittany K. Schwende, Will Huckins, Cory T Richards, Aled Rees, Charlotte W. Usselman, and Rachel N Lord

We are pleased to tell you that your paper has been accepted for publication in The Journal of Physiology.

Authors should note that it is too late at this point to offer corrections prior to proofing. Major corrections at proof stage, such as changes to figures, will be referred to the Editors for approval before they can be incorporated. Only minor changes, such as to style and consistency, should be made at proof stage. Changes that need to be made after proof stage will usually require a formal correction notice.

Yours sincerely,

Laura Bennet
Senior Editor
The Journal of Physiology

P.S. - You can help your research get the attention it deserves! Check out Wiley's free Promotion Guide for best-practice recommendations for promoting your work at www.wileyauthors.com/eeo/guide. You can learn more about Wiley Editing Services which offers professional video, design, and writing services to create shareable video abstracts, infographics, conference posters, lay summaries, and research news stories for your research at www.wileyauthors.com/eeo/promotion.

IMPORTANT NOTICE ABOUT OPEN ACCESS: To assist authors whose funding agencies mandate public access to published research findings sooner than 12 months after publication, The Journal of Physiology allows authors to pay an Open Access (OA) fee to have their papers made freely available immediately on publication.

You can check if your funder or institution has a Wiley Open Access Account here: <https://authorservices.wiley.com/author-resources/Journal-Authors/licensing-and-open-access/open-access/author-compliance-tool.html>.

EDITOR COMMENTS

Reviewing Editor:

Thank you for your careful revisions which have improved the clarity and flow of the manuscript. Congratulations on an excellent review, and thank you for your contribution to the Journal of Physiology.

REFeree COMMENTS

Referee #1:

Thank you for your thoughtful responses to my review. I have no further comments.

Referee #2:

The authors have done a great job responding to reviewer comments.